# Bone Turnover Alterations after Completing a Multistage Ultra-Trail: A Case Study

**DOI:** 10.3390/healthcare10050798

**Published:** 2022-04-25

**Authors:** Carlos Castellar-Otín, Miguel Lecina, Francisco Pradas

**Affiliations:** 1ENFYRED Research Group, Faculty of Health and Sports, University of Zaragoza, 22002 Huesca, Spain; castella@unizar.es (C.C.-O.); franprad@unizar.es (F.P.); 2Faculty of Health and Sports, University of Zaragoza, 22002 Huesca, Spain

**Keywords:** ultra-endurance, bone mass density, bone remodelling markers, bone formation, bone resorption

## Abstract

A series of case studies aimed to assess bone and stress fractures in a 768-km ultra-trail race for 11 days. Four nonprofessional male athletes completed the event without diagnosing any stress fracture. Bone turnover markers (osteocalcin (OC), serum C-terminal cross-linking telopeptide of type I collagen (CTX), bone-specific alkaline phosphatase (BALP), and serum turnover calcium (Ca^2+^)) were assessed before (pre) and after the race (post) and on days two and nine during the recovery period (rec2 and rec9), respectively. Results showed: post-pre-OC = −45.78%, BALP = −61.74%, CTX = +37.28% and Ca^2+^ = −3.60%. At rec2 and rec9, the four parameters did not return to their pre-run levels: OC, −48.31%; BALP, −61.66%; CTX, +11.93% and Ca^2+^, −3.38%; and OC = −25.12%, BALP = −54.65%, CTX = +93.41% and Ca^2+^ = +3.15%), respectively. Our results indicated that the ultra-trail race induced several changes in bone turnover markers, uncoupling of bone metabolism, increased bone resorption: OC and BALP and suppressed bone formation: CTX and Ca^2+^. Bone turnover markers can help determine the response of bone to extreme effort and might also help predict the risk of stress fractures.

## 1. Introduction

Regular physical activity has been recognized to have health benefits in general and specifically in the musculoskeletal system, increasing muscle strength and bone mineral density (BMD) [1]. Regular physical activity prevents multiple bone diseases, such as osteopenia or osteoporosis by increasing BMD [2]. Regardless of the importance of exercise in maintaining bone health, there is still no consensus in the scientific literature regarding the volume and intensity of effort required to prevent bone damage [3]. If exercise does not exert a minimum load on bone tissue, it will not increase BMD, as has been proven in studies carried out on swimmers [4]. Benefits in BMD depend on the type of exercise undertaken; weight-bearing exercise increases BMD, particularly at load-bearing sites, independent of muscular activity alone [5]. In contrast, when exercise overloads bone tissue because of excessive force, it may result in stress fractures and weaken BMD [6].

Among all the weight-bearing endurance sports, ultra-trail races include the longest (e.g., any distance in excess of the standard marathon distance 42.195 km or at least 6 h of duration) [7] and a great amount of negative and positive accumulated elevation, which increase the mechanical stress and consequently overload the microstructure of bone tissue (especially lower limbs) [8]. In addition, multi-stage ultra-trail competitions are usually held in extreme environments lasting several days and, as a consequence, athletes must carry their own provisions, resulting in additional weight and increasing the stress on bone tissue and reducing BMD [9]. Despite all these characteristics, the number of stress fractures found in competitions is relatively low [10,11]. Hoffman et al., in a descriptive study including 1212 ultra-runners, who completed an ultra-trail race, only found 0.3% of stress hip fractures and 1.9% of stress fractures involving tibia or fibula [12]. However, Scheer et al. found a higher incidence when assessing the stress fractures in a 12-month following study (10.3%) [13]. Apart from the length of the race, these runners have to complete a high volume of training prior to the competition in order to acquire the physiological and biomechanical adaptations required to face this extreme effort. Average training loads are between 66-83 km/week in adults and around 57 km/week in youth athletes [14]. This great amount of training may result in overtraining syndrome and subsequently in BMD loss increasing the likelihood of suffering stress fractures [15]. Thus, predictive markers that reflect stress of bone are needed to prevent stress fractures and help runners and coaches plan their training routines effectively [16].

BMD is usually assessed by using Dual-energy X-ray absorptiometry (DEXA) at the femur and, in the current criteria, OP is defined as a BMD T-score of −2.5 or lower at any one location or presenting a previous fragility fracture [4]. However, BMD analysis is not always conclusive as a predictive factor of stress fractures in the sports field [3]. Due to this fact, many researchers have raised the importance of bone turnover (BT) by analysing bone turnover markers (BTMs), as an essential factor when assessing bone microarchitecture and, therefore, the state of bone tissue [15,17]. BT consists of two dynamic processes: bone formation (anabolism) regulated by osteoblasts (responsible for bone matrix synthesis) and resorption (catabolism) mediated by osteoclasts (responsible for the secretion of proteolytic enzymes which digest bone matrix) [18]. BTMs are biochemical products measured usually in blood or urine that reflect the metabolic activity of bone, but which themselves have no function in controlling skeletal metabolism [19]. They are traditionally categorized as markers of bone formation or bone resorption. It has been shown that during the practice of endurance exercise there is an increase in markers related to bone resorption (serum C-terminal cross-linking telopeptide of type I collagen (CTX)) [20,21] and a decrease in those of formation (osteocalcin (OC), alkaline phosphatase (AP) and serum procollagen type I N propeptide (s-PINP)) causing a net loss of bone [22,23,24]. The analysis of BTMs offer some advantages over DXA alone: they allow the assessment of bone metabolic activity at a specific time, there are a high number of selective markers and the techniques are easily applicable and minimally invasive [25]. The analysis of BTMs, therefore, allows the monitoring of the likelihood of suffering stress fractures during the practice of physical activity by using selective biomarkers, apart from BMD [26]. The relationship between stress fractures and BT seems predisposed by an acceleration in the destruction of bone tissue that precedes the remodelling phase, which may cause a weakening of the tissue during this period and, therefore, increased probability of suffering stress fractures [27].

The aim of this case study was to assess the alterations suffered by four runners after completing a 768 km extreme ultra-trail race on bone turnover markers. We hypothesized that bone resorption markers (CTX and Ca^2+^) would increase and bone formation markers (CTX and BALP) would decrease. The alteration suffered in BTM may persist for days after the activity; a fact that may result in an increase in the suffering of stress fractures.

## 2. Case Report

Four non-professional healthy ultra-runners (38.08 ± 4.11 years) accepted participation in this case report after being invited by email. The four subjects included were males with broad experience (5 ± 1.26 years), well trained (11.61 ± 2.22 h·week^−1^) and had accumulated large amounts of elevation both positive and negative in the preparatory period (116,615 ± 37,462 m). It took them 154 h 43 min (SD ± 23 min) (equivalent to 51% of VO_2max_) to complete the 11 stages. They ran at an average speed of 5.11 ± 0.46 km h^−1^ and an average pace of 11 min 46 s (SD ± 3 min 4 s). No runner suffered any stress fracture. All participants were non-smokers and were not receiving medical, pharmacological, or dietary treatment.

Body composition measurements included: height, weight, skin folds and body mass index. All subjects were measured 2 h prior to the start of the race. Height measurement was made to the nearest 0.1 cm using a wall-mounted stadiometer (Seca 220, Seca, Hamburg, Germany), body weight was measured barefoot to the nearest 0.01 kg on calibrated electronic digital scales (Seca 769, Seca, Hamburg, Germany), skin folds used a compass accurate to ±0.2 mm (Seca 212, Seca, Hamburg, Germany) and a tape with an accuracy of ±1 mm was employed. Six skin folds were taken: abdominal, suprailiac, subscapular, tricipital, thigh and leg and perimeters; arms and legs were in a relaxed 90° position. The equations of Yushaz were used to calculate the percentage of fat [28] and the equation according to Lee to determine the percentage of muscle [29].

A cardiopulmonary test assessed the following physiological outcomes: maximum oxygen consumption (VO_2max_), heart rate maximum (HR_max_) and maximal aerobic speed (MAS). The laboratory test was performed on a treadmill (Pulsar, h/p/cosmos^®^, Nussdorf, Germany). The test was run on a 1% slope and the start speed was set to 8 km h^−1^, which increased 1 km·h^−1^ every minute. To warm up the subjects ran for 5 min on the treadmill operating at a speed of 6 km·h^−1^. Respired gases were collected with an Oxycon Proanalyzer (Erich Jaeger GmbH, Hoechberg, Germany). The gas analysis system was calibrated according to ambient temperature and humidity, air flow and VO_2_ and VCO_2_ concentrations. A pulsometer was used to evaluate the maximal heart rate (Vantage M, Polar, Finland). The participants’ pre-race characteristics are listed in Table 1.

The GR-11 route joins the Mediterranean and Atlantic coasts along the Pyrenees, covering 786 km in 11 stages. The average stage/day consisted of 71.49 km (SD ± 8.2), the average positive elevation was 4260.45 ± 1063.26, and the average negative elevation was 4258.63 ± 989.13. The race had a warm temperature, with values ranging from 13.1 to 17.6 °C, and the humidity was (60.1–70.9%). In-race hydration was provided ad libitum. The characteristics of the ultra-trail race are listed in Table 2. This table has been previously published [30]. This case report is part of a series of case studies aimed at studying the effects on runners’ health after completing this unique ultra-trail challenge called “GR-11”; accordingly, the characteristics of the ultra-trail (i.e., duration, positive and negative elevation) are the same in both case studies.

Twenty milliliters of venous blood (antecubital vein) were withdrawn from each participant pre- and post-race, rec2, and rec9 evaluations (90 min before and 10 min after finishing the race, two days and nine days in the morning). Blood samples were collected in two 5-mL Vacutainer tubes (Beliver Industrial State, Plymouth PL6 7BP, UK) without anticoagulant for serum isolation and in two 5-mL tubes containing ethylenediaminetetraacetic acid (EDTA) as an anticoagulant. Once collected, the blood samples were coagulated for 25-30 min at room temperature and then centrifuged at 2500 rpm for 10 min to remove the clots. Serum samples were aliquoted into Eppendorf tubes (Eppendorf AG, Hamburg, Germany), washed with diluted nitric acid, and stored at −80 °C until biochemical analysis. To facilitate the interpretation of the data, the change in analytical parameters was measured as follows: post-race, 2, 9 days less pre-race respectively. Statistical analyses were carried out using the Statistical Package for The Social Sciences software (IBM SPSS Statistics for Windows, version 26.0, 64 bits Edition, IBM Corp., Armonk, NY, USA). Descriptive analysis was carried out on all variables, and average, median and standard deviations were calculated. Normal distribution of the variables was verified by using Kolmogorov-Smirnov and Shapiro-Wilk tests, but normality criteria were not met because of the low number of subjects. *p*-value was calculated, but due to low number of subjects included and the design of the study as a series of case studies, its value was not considered for final analysis. The BTM changes analyzed are listed in Table 3. Range values were expressed for OC, ALP, CTX and Ca^2+^ according to age, sex, and race [31]. All bone formation markers included (OC and ALP) decreased their values when comparing pre- and post-exercise. Conversely, all the bone resorption markers decreased after race completion. During the recovery period, OC and ALP values remained above the basal line, even at rec9 (OC = −45.78% and BALP = − 54.65%). In contrast, the CTX values increased slightly at rec2 (+11.93%) but soared at rec9, with values close to +100%. (CTX = +93.41%). Serum calcium levels decreased slightly when comparing pre- vs. post-and pre- vs. rec2. However, the rec9 values exceeded the pre-race levels (Ca^2+^ = +3.15%) (See Table 3). The chronological sequence of BTMs is fully shown in four different figures included in Figure 1.

## 3. Discussion

The objective of this case report was to assess the alterations suffered on BTMs after completing a multi-stage ultra-trail and in the recovery period. The main finding of our study was suppression in bone formation and an increase in the bone resorption process, not only after completing the race but also in the recovery period (2 and 9 days after), respectively. To the best of the authors’ knowledge, this has been the first study to assess BTMs in such an extreme multi-stage ultra-trail after finishing the race and even nine days after in the recovery period. Only three previous studies have studied BTMs in ultra-endurance races so far, but the duration (from 245 to 308 km) and the elevation of these events were not as extreme as in the race included in this study [23,24,32]. To better understand the discussion, the text contains different points listed below.

### 3.1. Bone Formation Biomarkers

The results included in our study showed that bone formation markers reduced their values after completing an ultra-trail race competition and two and nine days after finishing it (See Table 3). These results fall in line with those previously reported by other authors [23,24,32]. However, these previous studies did not include such a long race nor assessed BTMs after nine days after completing the event. Despite the increase of new BTMs in recent years (e.g., Procollagen type I N-terminal propeptide (PINP) and procollagen type I C-terminal propeptide (PICP) [21], OC and BALP are still reliable markers of bone formation. Many of the most prestigious institutions related to bone health, including The National Health Allegiance (NBHA) [33], the International Osteoporosis Foundation (IOF) and the International Federation of Clinical Chemistry and Laboratory Medicine (IFCC), support the use of OC and BALP as clinical biomarkers for OP [19,34].

OC is a non-collagen protein synthesized exclusively by the osteoblasts and plays a pivotal role in osteogenesis [34]. Several studies have analyzed OC as the main BTM of bone resorption in ultramarathons [20,23,24,35]. Three out of the four found decreases in OC levels [20,23,24] but only one of them found no differences after finishing the race [32]. OC decrease during strenuous exercise has been explained by an increase in parathyroid hormone and cortisol [24]. The action of these hormones suppresses the activity of the osteoblasts or reduces osteoblast release as a consequence. Malm et al. found a four-fold increase in cortisol levels after completing a marathon [35]. In the same line, Knectle et al. found rises in cortisol and catecholamines and decreases in growth hormone, which shows the complexity of the alterations suffered by the hypothalamic-pituitary axis in these efforts [36]. In the recovery period, our study showed a relevant decrease of OC levels, even nine days after, which implies bone formation function remained partially suppressed days after completing the ultra-trail race. Other studies have also evaluated the activity of OC and have found similar results. Nizet et al. in a study that evaluated a marathon race, observed a decline in OC levels after the race (from 4.9 to 3.9 g/liter, −20%) and three days later. BALP is a homodimer anchored to the membrane of osteoblasts and matrix vesicles [37]. Although its exact function is not completely clear, the presence of alkaline phosphatase on the cell membrane is required for bone mineralization [38].

The results found in our study support the idea that repeated weight-bearing exercise may result in a suppression of the activity of BALP. However, BALP is mainly affected by hormonal levels altering its release. Malm et al., in a comparative study, only found significant decreases of BALP in the female group [39]. In this sense, the validity of BALP as a conclusive bone formation biomarker seems lower than OC or PINP [33]. Our study found higher decreases of BALP in the three measurements (−61.74%, −61.66% and −54.51%), despite the subjects of our study only being of male gender. This fact may be due to the excessive distance and the extraordinary elevation of the ultra-trail race in our study in comparison with previous research.

### 3.2. Bone Resorption Biomarkers

CTX is the result of osteoclastic bone resorption, and it is a type I breakdown product [38]. CTX has been proposed as the gold standard for assessing the bone resorption process [33]. Our study showed relevant decreases in CTX values after finishing the race and in the recovery period. Similarly, previous studies on marathons [40] and on ultra-trail races [20] have shown CTX increases ranging from +8% to +19%. The values found in our study were higher, especially at rec9 (+93.41%). The duration of the effort has been proposed as the main factor responsible for the increase of CTX [20]. According to many authors, [22,41] prolonged mechanical usage increases microdamage. It seems reasonable that higher values of CTX were found in our study because of the extreme duration of a 768 km run.

Ca^2+^ values slightly varied in the three measurements of our study (−3.60%, −3.38 and +3.15%). Similar increases were reported by Nila et al. [42]. After the race Ca^2+^ increases from 9.2 ± 0.1 (mean ± SE) to 9.8 ± 0.1 mg/dL (*p* < 0.01). Another study that evaluated the alterations of Ca^2+^ in marathon runners (eleven men and seven women) found no increases in Ca^2+^ after the end of the race. The activity of Ca^2+^ has also been associated with duration and intensity of effort, apart from the adrenergic activation post-exercise [36]. The multi-stage here studied (768 km and 11 stages) exceeds the duration and the elevation of the races analyzed in previous research, so the higher increases in Ca^2+^ values at post and rec2 are mainly justified by these specific characteristics.

### 3.3. Stress Fractures and BTMs

The four subjects included in our study suffered no stress injury in the course of the ultra-trail race despite the BTM alterations found in our study. The incidence of stress fractures that the scientific literature has reported in ultra-endurance sports and ultra-trail, in particular, is relevant, oscillating their values from 0.3% in femur or hip to 1.2% tibia or fibula [12]. High rates of bone remodelling have been associated with an increase in suffering of stress fractures, regardless of BMD loss measured by DEXA or ultrasound [38]. Tian et al., in a systematic review. found positive relationships between CTX and risk fractures (1.20, 95%CI, 1.05–1.37). The ACSM has shown that the most influential factors for stress fractures are exercise mode, intensity and duration. Accordingly, stress fractures occur as a result of excessive training activity due to repetitive mechanical loading [16]. It is because of these microfractures that bone resorption activity increases. Vasikaran et al., in a systematic review, found several studies that associated BTMS changes and subsequent fractures [19].

Traditionally, the studies investigating hip fractures have usually focused on women, due to hormonal reasons behind the development of OP and, consequent higher incidence of hip fractures. Nevertheless, studies including men have also found BTM changes prior to suffering a stress fracture [16]. Studies that have analyzed the relationship between BMD loss and the likelihood of stress fracture in running activities have found significant associations [16,27,36].

Bennell et al., in a 12-month prospective study, found no differences in OC values between athletes who suffered a stress fracture and did not (*p* = 0.010) [15]. On the contrary, Sayaka et al., in another study including young athletes, found an incidence of stress fractures higher than in other similar studies (11.4% of 316 athletes), but there was no statistical difference in BTMs [16]. We can conclude that the value of BTM alterations as a tool for predicting stress fractures is contradictory. This discrepancy in the results obtained is due to the variety of BTMs analyzed, as well as the different characteristics in the population included and differences in running activities studied. The characteristics of the ultra-trail races (i.e., duration and elevation) would require further investigation considering these characteristics.

### 3.4. Limitations

It must be considered that the sample size (*n* = 4) and the only male gender used in this study could be a limitation that had an impact on the results obtained. The main reason to justify the design of this study was due to the uniqueness and extreme conditions of the race (e.g., duration, number of stages and positive and negative elevation accumulated). As our study shows, there are several alterations on BTMs after finishing an ultra-trail race and at least nine days after in the recovery period. However, further investigation is still required in order to clarify the mechanisms involved in BT and its relation with BMD and/or BC loss, and, ultimately, the etiology of stress fractures in these efforts. More epidemiological studies, including analyses of BTMS and DXA or ultra sound measurements, are needed to better elucidate the mechanisms involved in BMD.

## 4. Conclusions

According to this study, during an ultra-trail race it appears that bone resorption is increased and, conversely, bone formation is suppressed, resulting in a transient uncoupling of BT. The levels of all BTMs analyzed remained altered when compared with pre-run levels, especially in CTX and OC, even nine days later. This study showed that a 768-km multi-stage ultra-trail induces changes in the OC/Ca^2+^/BALP/CTX interaction, which may result in an increase in the likelihood of stress fractures as a consequence of damage to bone tissue. The special preparation that these athletes had to carry out to face the race implies a great amount of training volume (e.g., kilometers accumulated, n° sessions·week^−1^) prior to the race, so the study of which BTMs reflect bone damage may help in preventing runners suffering from BMD loss and in avoidance of stress fractures. Considering the results of this case report, runners and coaches should analyze alterations in BTMs, not only immediately after the race but also in the recovery period. The analysis of the BTMs here presented offers valuable information about load in bone tissue during the training process and may help runners reduce the likelihood of suffering stress fractures. The training process of these races requires a structured program of scientific monitoring in physiological, biomechanical and performance areas. According to these findings, BTMs should be measured as part of the preparation routine for ultra-races to prevent ultra-runners suffering bone turnover alterations.

## Figures and Tables

**Figure 1 healthcare-10-00798-f001:**
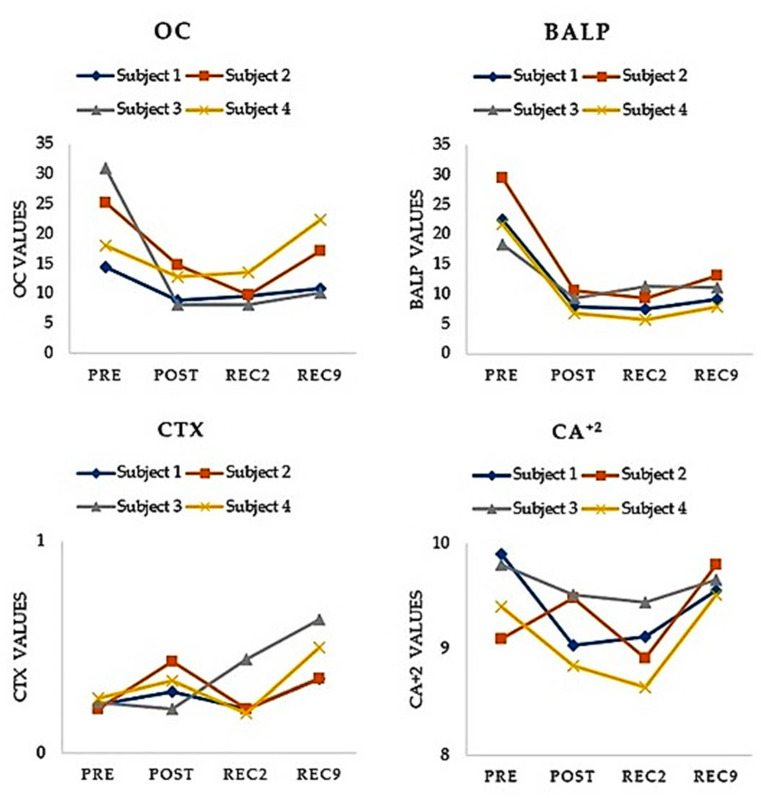
Chronological sequence of serum variables of bone metabolism of each of the 4 subjects. No statistically significant differences were found when comparing pre vs. post, rec2 and rec9 (*p* > 0.05).

**Table 1 healthcare-10-00798-t001:** Pre-race individual characteristics of the population included (*n* = 4).

Parameters	Subject 1	Subject 2	Subject 3	Subject 4
Age (years)	33	37	41	42
VO_2max_ (mL/kg/min^−1^)	58.28	70.6	67.1	50.71
HR_max_ (beats·min^−1^)	194	186	194	176
Maximal aerobic speed (km·h^−1^)	18	17	16.7	16
Height (cm)	180.7	176.1	172.3	173.9
Weight (kg)	79.1	64.9	60.8	77.3
BMI	24.2	20.9	20.5	25.6
Fat mass (%)	8.82	6.88	8.70	8.14
Muscle mass (%)	43.4	47.38	57.63	38.55
Experience (years)	6	6	4	7
Distance covered (h·week^−1^)	11	11	15	11
Annual slope accumulated (m)	140,655	120,404	156,000	70,000

BMI, body mass index; HRmax, heart rate maximum; VO_2max_: maximum oxygen consumption.

**Table 2 healthcare-10-00798-t002:** Characteristics of the extreme ultra-trail [30].

Stages	Distance (km)	Elevation (m+)	Elevation (m−)
1	78.5	3136	3024
2	72.3	3886	3458
3	72	4655	4044
4	68.1	5660	4581
5	72.6	5411	6336
6	76.1	5344	4788
7	63.7	5492	5163
8	66	3641	4576
9	66.1	3361	3841
10	66.5	2958	2934
11	83	3321	4100
Total	784.9	46,865	46,845
Md	71.35	4260.45	4258.63
Sd	±6.00	±1063.26	±989.13

**Table 3 healthcare-10-00798-t003:** Blood parameters before (baseline) and after race (post-exercise day 2 and post-exercise day 9).

Parameter Blood (Reference Values)	Before-Race	Post-Race		
	Pre (Baseline) Value	Post (Post-Exercise) Value (% Difference)	Day 2 (rec2) Value (% Difference)	Day 9 (rec9) Value (% Difference)
OC (ng/mL) (13.98–41.99)	22.20 ± 7.41	11.15 ± 3.14 ↓ (−45.78)	10.30 ± 2.29 ↓ (−48.31)	15.14 ± 5.73 ↓ (−25.12)
BALP (ug/L) (6–30)	23.03 ± 4.68	8.64 ± 1.63 ↓ (−61.74)	8.50 ± 2.37 ↓ (−61.66)	10.29 ± 2.30 ↓ (−54.65)
CTX (μg/L) (0.23–0.94)	0.24 ± 0.02	0.32 ± 0.09 ↑ (+37.28)	0.26 ± 0.12 ↑ (+11.93)	0.46 ± 0.14 ↑ (+93.41)
Ca^2+^ (mg/L) (8.70–10.40)	9.35 ± 0.33	9.22 ± 0.32 ↓ (−3.60)	9.03 ± 0.34 ↓ (−3.38)	9.64 ± 0.12 ↑ (+3.15)

Data are expressed as absolute values and as ± percentages from baseline values. OC, osteocalcin; BALP, alkaline phosphatase; CTX, C-terminal cross-linking telopeptide of type I collagen; Ca^2+^, calcium.

## Data Availability

Information about the case report is available at http://gr11en11.org/ (accessed on 26 October 2021).

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
