# Peer review of "Bone Turnover Alterations after Completing a Multistage Ultra-Trail: A Case Study"

_healthcare, 2022, doi:10.3390/healthcare10050798_

Round 1
Reviewer 1 Report
Your Case Study on these 4 participants may help open doors for more research on the topic of stress fractures.
In the Introduction, as you discuss the role of physical activity on bone health, please be sure to discuss the significance of weight-bearing activities. You mentioned "weight-bearing" in the Discussion, but did not mention the word "weight-bearing" in the Introduction, so please be sure to include it. Also, bone mineral content has the acronym "BMC" and not "BC"
Your reports used blood measurements and parameters. For future research, you could mention also using urine and sweat measurements, as studies have found that excessive calcium can be lost in both urine and sweat from high-volume and/or high-intensity exercise.
Participants were non-smokers, which is important as smoking can be detrimental to bone health. But so is excessive alcohol consumption, and future studies should ask this question.
Your results showed that bone formation markers were still reduced 9 days after the ultra-trail race competition (and other markers such as cortisol increasing). It would have been great to continue to measure bone formation markers for longer than 9 days afterwards, until as long as it takes until they go back to normal levels. For example, does it take 2, 3 or 4 full weeks or longer? This would be helpful in knowing exactly how long it takes the body to fully recover, and has implications for exercise training and recovery to prevent overtraining and stress fractures. This should be noted for future research.
Author Response
Response to Reviewer 1 Comments
Dear reviewer 1,
We appreciate your review. We feel proud that you have found it worth reviewing our series of case studies. The results found in this study warn runners of the consequences of running this extreme competition. The alterations found continued even nine days after completing the race. Therefore, this study shows some biomarkers that could help monitor the load of bone tissue and prevent runners from increasing the likelihood of suffering stress fractures.
Only three studies (Mouzopoulos et al., 2007; Kerschan-Schindl et al., 2009; Shin et al., 2012) have focused on bone damage and bone turnover in this sport, but all of them did not include such a long race or accumulated elevation. As accumulated elevation and duration of the weight-bearing exercise have been proposed as factors for altering bone turnover markers, we designed an exploratory case study to confirm these hypotheses.
Here is a point-by-point response to your main notes and concerns. We have highlighted the changes within the manuscript in yellow colour and a comment has been added in the original document.
Comment 1. “In the Introduction, as you discuss the role of physical activity on bone health, please be sure to discuss the significance of weight-bearing activities. You mentioned "weight-bearing" in the Discussion, but did not mention the word "weight-bearing" in the Introduction, so please be sure to include it. Also, bone mineral content has the acronym "BMC" and not "BC"
Response 1. We fully agree. Consequently, we have added a comparison between non-weight-bearing activities like swimming or cycling and ultra-trail races. Additionally, we have added physiological effects on bone mineral density.
Comment 2. Your reports used blood measurements and parameters. For future research, you could mention also using urine and sweat measurements, as studies have found that excessive calcium can be lost in both urine and sweat from high-volume and/or high-intensity exercise.
Participants were non-smokers, which is important as smoking can be detrimental to bone health. But so is excessive alcohol consumption, and future studies should ask this question.
Response 2. We appreciate your suggestions and we are going to take them into account for future investigations.
Comment 3. “Your results showed that bone formation markers were still reduced 9 days after the ultra-trail race competition (and other markers such as cortisol increasing). It would have been great to continue to measure bone formation markers for longer than 9 days afterwards, until as long as it takes until they go back to normal levels. For example, does it take 2, 3 or 4 full weeks or longer? This would be helpful in knowing exactly how long it takes the body to fully recover and has implications for exercise training and recovery to prevent overtraining and stress fractures. This should be noted for future research.”
Response 3. We support your suggestion. We support your suggestion. We are planning new investigations to expand these findings and we would like to analyze bone turnover biomarkers in the long term.
Yours faithfully,
Miguel Lecina
University of Zaragoza

Reviewer 2 Report
This case report describes four athletes undergoing systematic measurment of bone turnover markers before and after a long-trail run. The authors need to greatly improve the manuscript in terms of methodology.
Major revisions:
Table 1: please avoid describing means and SD for 4 subjects, instead, present each subject and their characteristics.
Table 4: there is not any significant correlations, the legend is misleading (all P values are > 0.05). Please delete this table. Describe that no statistically significant correlations were found, probably due to small sample size.
Remove lines 130 through 142, because no correlations were found. Instead, it might be valuable drawing a figure with each single bone turnover marker before and after the run, for each subject.
Rewrite the discussion taking into account that correlations in four subjects cannot be established. Please focus on bone turnover markers and their significance for prediction of stress fractures only.
Abstract: line 20 bone formation and resorption markers were wrongly cited in parentheses
Title: delete 'do not imply stress fractures' because your data do not support that.
Conclusions: please delete line 258 through line 264, because this is referred to correlations
Author Response
Dear reviewer 2,
Thank you for allowing us to submit a revised draft of our manuscript titled “Bone turnover alterations after completing a Multistage Ultra-Trail: A Case Study.” to "Healthcare".
We appreciate the time and effort that you have dedicated to providing your valuable feedback on our manuscript. Consequently, we have been able to incorporate changes to reflect most of the comments provided by you. We have highlighted the changes within the manuscript in green colour and a comment has been added to the original document.
Here is a point-by-point response to your main notes and concerns. Additionally following the manuscript that you gently attached in the review process we have corrected minor issues as misspellings or format corrections.
- Comment 1. “Table 1: please avoid describing means and SD for 4 subjects, instead, present each subject and its characteristics.”
- Response 1. Table 1 has been changed and now shows the characteristics of each subject individually. Means and sd have been deleted.
- Comment 2. “Table 4: there is not any significant correlations, the legend is misleading (all P values are > 0.05). Please delete this table. Describe that no statistically significant correlations were found, probably due to small sample size.”
- Response 2. We agree. Table 4 has been deleted. We have added in lines 129-133 the statistical analysis performed and the reason why the p-value was not statistically significant.
- Comment 3. “Remove lines 130 through 142, because no correlations were found. Instead, it might be valuable drawing a figure with each single bone turnover marker before and after the run, for each subject.”
- Response 3. Lines removed. This figure shows the chronological bone turnover biomarkers alterations of each subject in the four different measurements.
- Comment 4. “Rewrite the discussion taking into account that correlations in four subjects cannot be established. Please focus on bone turnover markers and their significance for prediction of stress fractures only.”
- Response 4. We agree. We have rewritten the discussion adding a new point solely focused on stress fractures and bone turnover biomarkers changes.
- Comment 5. “Abstract: line 20 bone formation and resorption markers were wrongly cited in parentheses”
- Response 5. Solved
- Comment 6. “·Title: delete 'do not imply stress fractures' because your data do not support that.”
- Response 6. The title has been rewritten following your suggestion.
- Comment 7. “Conclusions: please delete line 258 through line 264, because this is referred to as correlations”
- Response 7. Every paragraph related to associations between training parameters and Bone turnover biomarkers has been deleted.
Yours faithfully,
Miguel Lecina
University of Zaragoza

Reviewer 3 Report
Comment to the manuscript ID: healthcare-1649497entitled Bone turnover alterations do not imply stress fractures after completing a 768-Kilometer Multistage Ultra-Trail: A Case Study.
Your research has a HIGH Similarity with a previous manuscript: "Bone turnover alterations do not imply stress fractures after completing a 768-Kilometer Multistage Ultra-Trail: A Case Study."
I recommend authors rewrite the manuscript.
Despite this, I would like to know how calculate the sample size and I want to know if your subjects are the same as the previous research.
Authors should add the statistical analysis and write a NEW manuscript.
Author Response
Response to Reviewer 3 Comments
Dear reviewer 3,
Thank you for allowing us to submit a revised draft of our manuscript titled “Bone turnover alterations after completing a Multistage Ultra-Trail: A Case Study.” to "Healthcare".
We appreciate the time and effort that you have dedicated to providing your valuable feedback on our manuscript. As a consequence, we have been able to incorporate changes to reflect some of the comments provided by you. Moreover, we have highlighted the changes within the manuscript in fuchsia colour and a note has been written in the original document for each of your suggestions. Here is a point-by-point response to your main notes and concerns.
Comment 1:” Your research has a HIGH Similarity with a previous manuscript: "Bone turnover alterations do not imply stress fractures after completing a 768-Kilometer Multistage Ultra-Trail: A Case Study."
Response 1: The title you refer it is the same as this study. Please revise what title are you referring to. This manuscript is new as well as the results and data have not been published in any other journal. Therefore, we suppose you must refer to this one as “768-km Multi-Stage Ultra-Trail Case Study-Muscle Damage, Biochemical Alterations and Strength Loss on Lower Limbs” (Lecina et al., 2022). That study was utterly different despite the race being the same. The main goal of that study was to assess strength losses in lower limbs and evaluate rhabdomyolysis. Muscle damage is not related to bone damage. The assessment tools and procedures likewise the mechanisms involved in the aetiology are not linked, so a different study is required. Many studies are based on the same population but provided the results and the discussion are different does not imply any ethical issue and does not interfere with the validity of the study (Lipman et al., 2014, 2016, 2021). So far, only three studies have assessed bone damage biomarkers in ultra-marathon races (Mouzopoulos et al., 2007; Kerschan-Schindl et al., 2009; Shin et al., 2012) and no one of them evaluated the duration of the alterations nine days after. Many studies are based on the same population but provided the results and the discussion are different does not imply any ethical issue and does not interfere with the validity of the study. So far, only three studies have assessed bone damage biomarkers in ultra-marathon races (Mouzopoulos et al., 2007; Kerschan-Schindl et al., 2009; Shin et al., 2012) no one of them evaluated the duration of the alterations nine days after. The lack of studies assessing bone turnover alterations has been described in many systematic and scoping reviews (Zaryski and Smith, 2005; Gabrielli et al., 2018; Knechtle and Nikolaidis, 2018; Rojas-Valverde et al., 2019) and it results in a matter of concern in the light of the incidence of stress fractures that these runners suffer.
Comment 2. “I recommend authors rewrite the manuscript.”
Response 2. We partially agree. As a consequence, we have rewritten more than 60% of the article following suggestions. However, we do not believe that the whole document must be rewritten after the changes we have carried out. With due respect we believe that you should consider other reviewers´ assessments who have assessed positively this case study.
Comment 3. “Despite this, I would like to know how calculate the sample size and I want to know if your subjects are the same as the previous research.”
Response 3. The uniqueness of the multi-stage ultra-trail offered an incredible opportunity to study the bone turnover alterations in such an extreme effort. The design of the study as a case study implies that due to the lower number of populations included p-values are not going to reach significance Even so, we performed this analysis that you can see in point 2.7 where a high-detail description was added
Comment 4. “Authors should add the statistical analysis and write a NEW manuscript.”
Response 4. A description of the statistical analysis performed has been added in lines 129 -134. We respect your point, but we believe that a new document is not required considering the other two reviewers ‘opinions. You must consider that this is a case study and is completely normal that the p-value is not statistically significant. Additionally, we performed a statistical analysis and we find very strong relations in many parameters, which encourages us to keep studying the relationship between bone damage biomarkers and training parameters. The statistical analysis performed was the following:
Statistical analyses were carried out using the Statistical Package for The Social Sciences software (IBM SPSS Statistics for Windows, version 26.0, 64 bits Edition, IBM Corp., Armonk, NY, USA). Descriptive analysis was carried out in all variables, and average, median and standard deviation were calculated. Normal distribution of the variables was verified by using Kolmogorov-Smirnov and Shapiro-Wilk tests, but normality criteria were not met because of the low number of subjects. Consequently, a non-parametric test was performed. Bivariate inferential analyses using Spearmen correlation were performed to contrast the association between the change of strength tests and analytical parameters and between these variables and those that measure the volume of training. The nonparametric correlation coefficient was applied because normality was not reliable. A confidence level of 95% was established, and Spearmen´s rank correlation (Rs) was used to de-scribe the relation between variables. Five different ranges were set according to Fowler et al. criteria to evaluate the strength between the strength variables and blood parameters [38]; (0.00 to 0.19) very weak, 0.20 to 0.39 weak, (0.40 to 0.69) moderate correlation; (0.70 to 0.89) strong; (0.90 to 1.00) very strong. P-value was calculated but due to low number of subjects included its value was not considered for final analysis. In order to facilitate the interpretation of data, the changes in OC, BALP and Ca+2 were measured as follows; post, rec2 and rec9 less pre. Conversely, CTX changes were measured inversely.
We appreciate your suggestions and do not hesitate that shall be considered them in future investigations. The sample size was not determined because it was conceived since its inception as an exploratory case study due to the uniqueness and extreme conditions of this race (i.e., 768 kilometers, eleven stages, accumulated negative and positive elevation).
We hope you may find convenient the information added in this email, and please do not hesitate to contact us regarding any queries you might have.
Yours faithfully,
Miguel Lecina Monge
University of Zaragoza.

Round 2
Reviewer 2 Report
- Comment 5. “Abstract: line 20 bone formation and resorption markers were wrongly cited in parentheses”
- Response 5. Solved
This is still unsolved. Formation is: BAP and OC. Resorption is: CTX and Ca++.
Please correct.
No further comments.
Author Response
Dear reviewer 2,
Thank you for allowing us to resubmit a revised draft of our manuscript titled “Bone turnover alterations after completing a Multistage Ultra-Trail: A Case Study.” to "Healthcare".
We appreciate the time and effort that you have dedicated again to providing your valuable feedback on our edited manuscript. Consequently, we have been able to incorporate the only change that you suggested. We have highlighted the changes within the manuscript in green colour and a comment has been added to the original document.
- Comment 5. “Abstract: line 20 bone formation and resorption markers were wrongly cited in parentheses”
- Response 5. Solved
Yours faithfully,
Miguel Lecina
University of Zaragoza
Reviewer 3 Report
Dear authors, changes required have not been improved. Your manuscript continues with a high plagiarism rate.
Changes realized in your manuscript are minor.
Authors have not answer properly to the question required:
how they calculate the sample size.? What is the difference between 1 and 100 runners?
Lines 70-79: The paragraph is the same as previous research with differents parameters.
Line 80: Case report is the same as the previous research. Authors have modified in text NOTHING.
The manuscript contains some repeated paragraphs.
Authors must rewrite the manuscript.
Author Response
Dear reviewer 3,
Thank you for allowing us to submit a new version of the manuscript titled “Bone turnover alterations after completing a Multistage Ultra-Trail: A Case Study.” to "Healthcare".
We appreciate the time and effort that you have dedicated to providing your valuable feedback on the new version of our manuscript. Unfortunately, we can not rewrite the whole document because all the other reviewers have already accepted the previous version of this case report. We believe that rewriting the whole document is far from reasonable.
We would like to consider your suggestion but you have not referred yet to the article that allegedly we have plagiarized, although we have politely asked previously. We previously explained why the three paragraphs were similar in their structure.
This study is included in a series of case studies that were approved by The Govern of Aragon and received funds. The protocol was the same for both articles so, the paragraphs which explain these points (i.e., blood extraction, the characteristics of the race and the four subjects were the same.). These paragraphs account for 591 words and the whole article is 5404 words. Are you still supporting the idea that this case report has been copied??? Please probe it and reference the article you mention. The introduction, the case report, the discussion and the references are different as we explained in the previous letter so, we can not understand your point.
Only one paragraph in the introduction contains the same structure (line 80) and has been rewritten following your suggestion.
Comment 2 “Changes realized in your manuscript are minor.”
Response 2: Changes realized in our manuscript have been considered major for the rest of the reviewers who have fully supported the acceptance of this article. We can not rewrite an article previously accepted by the other reviewers. Following your suggestions, we have replaced the lines 70-86 with a new paragraph.
Comment 3: Authors have not to answer properly the question required: how they calculate the sample size.? What is the difference between 1 and 100 runners?
The GR11 was not a competition like Comrades Marathon, it was a challenge specially selected in these case reports assessing the effect of this unique challenge (i.e., duration, elevation, stages) on muscle damage (Lecina et al., 2022) and bone damage turnover (NOT PUBLISHED ). As we refer to in the paragraph limitations (line 252), the low number of subjects included prevents us from concluding statistical differences (p-value > 0,05). In this case study, the controls and the cases are the same (n = 4). This is a pilot study only designed to explore safely future research properly designed as an article (double-blinded, comparative randomized prospective). As the whole population was assessed we could not match every runner with a pair-matched control.
In future investigations, we will use the following formula to calculate the size-effect
The formula is: nc = ne=Z α/2*Z β 2*S2d2, where d is the average of the individual differences between the basal and posterior values, S2 is the variance of both distributions, which are assumed to be equal. Z α/2 is the value of the axis of the abscissas of the standard normal function, where the probability of (1-α) accumulates for a bilateral hypothesis contrast and Z β is the value of the axis of the abscissas of the standard normal function, where the probability of (1-β accumulates). (García-garcía, Reding-bernal and López-alvarenga, 2013). This case report was solely designed to analyze statistical differences we should have included a number between 30 and 50 subjects (García-garcía, Reding-bernal and López-alvarenga, 2013).
We appreciate your comments and we will consider them for future investigations.
Yours faithfully,
Miguel Lecina
University of Zaragoza